# Cost-effectiveness analysis of an intervention project engaging Traditional and Religious Leaders to improve uptake of childhood immunization in southern Nigeria

**Angela E. Oyo-Ita**[1,2]*, **Patrick Hanlon**[3,4], **Ogonna Nwankwo**[1], **Xavier Bosch-Capblanch**[3,4], **Dachi Arikpo**[2], **Ekperonne Esu**[2,5], **Christian Auer**[3,4], **Martin Meremikwu**[2,6]

1 Department of Community Medicine, College of Medical Sciences, University of Calabar, Calabar, Nigeria, 2 Effective Health Care Alliance Programme, Institute of Tropical Disease, Research and Prevention, University of Calabar Teaching Hospital, Calabar, Nigeria, 3 Swiss Tropical and Public Health Institute, Basel, Switzerland, 4 University of Basel, Basel, Switzerland, 5 Department of Public Health, College of Medical Sciences, University of Calabar, Calabar, Nigeria, 6 Department of Paediatrics, College of Medical Sciences, University of Calabar, Calabar, Nigeria

* oyo_ita@yahoo.com

**Data Availability Statement:** This paper is based on an effectiveness study. The repository for the data is DOI:10.5061/dryad.gmsbcc2ms. Data for

## Abstract

Vaccination is a cost-effective public health intervention, yet evidence abounds that vaccination uptake is still poor in many low- and middle-income countries. Traditional and Religious Leaders play a substantial role in improving the uptake of health services such as immunization. However, there is paucity of evidence on the cost-effectiveness of using such strategies. This study aimed to assess the cost-effectiveness of using a multi-faceted intervention that included traditional and religious leaders for community engagement to improve uptake of routine immunisation services in communities in Cross River State, Southern Nigeria. The target population for the intervention was traditional and religious leaders in randomly selected communities in Cross River State. The impact of the intervention on the uptake of routine vaccination among children 0 to 23 months was assessed using a cluster randomized trials. Outcome assessments were performed at the end of the project (36 months). The cost of the intervention was obtained from the accounting records for expenditures incurred in the course of implementing the intervention. Costs were assessed from the health provider perspective. The cost-effectiveness analysis showed that the incremental cost of the initial implementation of the intervention was US$19,357and that the incremental effect was 323 measles cases averted, resulting in an incremental cost-effectiveness ratio (ICER) of US$60/measles case averted. However, for subsequent scale-up of the interventions to new areas not requiring a repeat expenditure of some of the initial capital expenditure the ICER was estimated to be US$34 per measles case averted. Involving the traditional and religious leaders in vaccination is a cost-effective strategy for improving the uptake of childhood routine vaccinations.

the cost effectiveness is as presented in the data provided in the paper.

**Funding:** This study was funded by International Initiative for Impact Evaluation (3Ie) [https://www.3ieimpact.org/]. The Funder played no role in the study design, data collection, and analysis, decision to publish, or preparation of the manuscript.

**Competing interests:** The authors have declared that no competing interests exist.

# Introduction

Vaccination is a cost-effective public health intervention [1, 2]. It is estimated to cost less than $100 to $1000 per disability-adjusted life year averted in low-and-middle-income countries [2]. Despite its benefits, vaccination uptake remains poor in many low- and middle- income countries [3]. For instance, a recent survey reported that slightly less than one-third (31%) of all children in Nigeria had received age-appropriate vaccines before the age of 24 months [4]. Apart from the weakness of the health system, family characteristics, parental attitudes and behaviours, limitations in immunization-related communication and information have been associated with under-vaccination and non-vaccination of children [3, 5]. A recent systematic review (6) found paucity of evidence on effectiveness of the interventions for improving vaccine coverage among children. Due to the multiplicity of the factors that are associated with under-vaccination and non-vaccination, multi-faceted local interventions may have promising beneficial effects [5].

Community participation for demand creation for healthcare is one of the bedrocks of primary healthcare [6]. Achieving effective community participation has been shown to be dependent on the employment of adequate numbers of formal community health workers, and the engagement of community members with full participation of their established strata of authorities especially those that serve as gate keepers [6]. Traditional and Religious Leaders (TRLs) are influential and respected in their communities as opinion formers and guides in religious, social and family life [7]. Thus, they often serve as gatekeepers in the community providing the link between the formal health system and the community. In addition, they can play an informal role in the implementation of health policies at communal levels [7]. In playing their role, they often engage in informing and educating their communities on pertinent health issues. Community gatekeepers have been portrayed to be beneficial in Nigeriaand they have played major role in influencing the uptake of Polio vaccination during campaigns in Northern states of Nigeria [8, 9].

Given that there is paucity of evidence on the cost effectiveness of community-driven strategies to improve the demand and uptake of immunization services, this study aimed to assess the cost-effectiveness of using traditional and religious leaders for community engagement in improving uptake of routine immunisation services in communities in Cross River State, Southern Nigeria. The findings will provide further evidence to decision-makers and government for better policy making and resource allocation in improving immunization uptakes.

# Methods

Ethical clearance was obtained from Cross River State Health Ethics Committee. The approval was conveyed in a letter referenced: RP/REC/2016/398. Oral consent was obtained from participants. The target population for the intervention was the traditional and religious leaders in randomly selected communities in Cross River State. The impact of the training on the uptake of routine vaccination among children 0 to 23 months was assessed. Childhood immunization service is the responsibility of the National and State Primary Health Care Agencies. These agencies support training, logistics, and policies for routine vaccination while the implementation of the services is the direct responsibility of the Local Government Councils. Decision-making on immunization involves these 3-tier structures operating at the National, State and Local Government levels. The study perspective was from a healthcare payer system but we considered only the payment from the provider's angle. We considered that the cost for the service users in both arms of the trial would be the same because no intervention targeted the service users directly.

## Study design

The study was a cluster-randomized controlled trial in southern Nigeria. Ethical clearance for the study was obtained from the Cross River State Ministry of Health Research Committee. Eight of the 18 Local Government Areas (LGAs) in Cross River State, Nigeria were randomly selected with 4 as intervention and 4 as control LGAs. Three wards were randomly selected from each LGA, and 4 villages from each ward making a total of 48 villages per study arm. As it was not feasible to blind participants, survey respondents and data collectors were blinded. In addition, non-contiguous villages were selected to reduce the risk of contamination.

The sample size assumed the pre-intervention proportion of fully vaccinated infants (0–23 months) was 53% and wished to detect a 10% increase post-intervention) with 80% power, a 5% significance level and k = 0.18. Further details about the sample size calculation, randomization and blinding procedures are in the report of a published clinical trial with registration number: PACTR202008784222254 [10].

The intervention had several components which were designed to fit the structure of the Primary Health Care system that has the Ward Development Committee (WDC) headed by a traditional leader executing its oversight function. Its main focus included stakeholder engagement and the training of traditional and religious leaders in the intervention arm; the control arm had only the routine care. The components of the intervention are as described in Table 1 below:

The sessions were interactive and participatory. Methods of training adopted included brainstorming, large and small group discussions, role-plays, problem-solving case studies, and use of learning aids. The time horizon for the study from the inception to the end-line assessment was from March, 2016 to February, 2019 corresponding to three years. This enabled us to use the same time horizon for evaluating both the effect and the cost-effectiveness of the intervention in keeping with recommendations by Ramsey et al. for studies carried out on an intervention basis [11, 12]. All discount rates which were drawn from the Nigerian Central Bank Treasury Bill rate at the end of each year were applied to both the costs and outcomes. This was in keeping with a recommendation from Drummond et al [13]. The rates were 13.97%, 13.01% and 10.91% for 2016, 2017 and 2018 respectively [14–16].

The outcomes measured were the uptake of vaccination as shown by the number of children who had received vaccine antigens and the time at which the vaccine was received relative to the recommended schedule. The uptake of vaccination was measured through a triangulation of data obtained from an interviewer-administered questionnaire adapted from the WHO vaccination coverage tool, children's immunization history obtained from their vaccination

**Table 1. Component activities for the intervention.**

| Components of intervention | Description of input |
| --- | --- |
| 1.Engagement and training of Traditional and Religious Leaders(TRLs) | Eight rounds of training were conducted within 18 months for 97 TRLs. Training focused on the role of leaders in identifying challenges in routine vaccination, transformational leadership, good communication skills, what vaccination is. It aimed at improving the TRLs' leadership role in the community and the WDC. |
| 2.Community engagement | Routine community engagements were used by the TRLs to share information on routine immunization as a means of encouraging community members to get their children vaccinated. |
| 3.Health service | Health workers were trained for one day to provide user-friendly data generated from routine immunization to the TRLs. |
| 4. Strengthening of the WDC | It was ensured that the WDC meetings were held routinely to facilitate the interaction of the members with the TRLs. |

**Table 2. Net effect of the intervention in selected outcomes.**

| | Control | | Intervention | | P value | Population (as if intervention) | Control | | Intervention | | Net effect | Discounted to 2019 |
|---|---|---|---|---|---|---|---|---|---|---|---|---|
| | Before | After | Before | After | | | | | | | | |
| | N(%) | N (%) | N (%) | N(%) | | | Effect % | Effect | Effect % | Effect | | |
| Not vaccinated | 125 (9.6) | 128 (10.0) | 87 (6.7) | 5 (0.4) | <0.001 | 1,250 | 0 | 0 | -7 | -83 | -83 | -104 |
| Partial | 551 (42.4) | 449 (35.2) | 619 (47.7) | 610 (47.8) | 0.69 | 1,271 | -7 | -89 | 0 | 0 | 89 | 112 |
| Penta 1 | 531 (46.3) | 496 (43.0) | 511 (46.1) | 694 (59.5) | <0.001 | 1,157 | -3 | -35 | 14 | 162 | 197 | 247 |
| Penta 3 | 273 (27.9) | 243 (24.1) | 226 (23.5) | 292 (29.4) | 0.005 | 1,007 | -4 | -40 | 5 | 50 | 91 | 114 |
| measles | 155 (23.9) | 154 (24.4) | 124 (19.1 | 240 (41.0) | <0.001 | 585 | 0 | 0 | 22 | 129 | 129 | 162 |

cards and routine data from the DHIS [10]. The definition of timely vaccination allowed for no more than two weeks after the scheduled vaccination date. The intervention was effective in reducing the proportion of unvaccinated children from 7% to 0.4% (p = 0.001) and improving timeliness of Penta 3 (OR 1.55; 95% CI: 1.14, 2.12; p = 0.005) and Measles (OR 2.81; 96% CI: 1.93–4.1; p<0.001) vaccination (Table 2).

Some of the effect parameters derived from the study, including a calculation of the counterfactual, which would have resulted, based on the results of the control arm of the study are shown in Table 2. The counterfactuals were then used to calculate the net effect on the intervention arm. The counterfactual represents here the results, which would have been attained in any case, if no intervention had taken place. The percentage changes in the control arm between baseline and end-line were applied to the end-line 'intervention' population to derive the counterfactual effect in numbers. Then, the counterfactual effect in numbers was deducted from the effect derived in the intervention population to obtain 'net' effect. Both the calculated net effect and the net cost were discounted to 2019 according to recommendations from Drummond and colleagues [13].

## Cost estimation

The cost of the intervention was obtained from the accounting records for expenditures incurred in the course of implementing the intervention. The expenditures covered administrative and interventional costs such as advocacy visits, developing and printing of the training tools, staff training, implementation, monitoring, and user cost encompassing the estimate cost of the time the TRLs would spend in providing the intervention if they were to be employed. The costing was done based on the J-Pal costing guideline [17]. The cost perspective is that of the intervention implementers who are assumed to be the local government agencies. Additionally, the opportunity cost of the TRL (time/days time spent for the intervention) was converted to salary costs for those days, given that these are costs that the provider may also have to cover. The costs covered the period from the inception of the project in March, 2016 to the final data collection in February, 2019. The exchange rate used was NGN306.30 to a US Dollar being the "Central Rate" exchange rate used by the Central Bank of Nigeria [18].

The details of the total costs, as well as the marginal intervention running (variable) costs excluding fixed costs of start-up investments, can be seen in Table 3. The marginal costs are the variable costs excluding the pre-testing of the training tool, the consultancy services to

**Table 3. Detail of costs under two scenarios: full costs and intervention running costs only.**

| | | Scenario 1 | Scenario 2 |
|---|---|---|---|
| | **Basic cost collection template** | **Total costs NGN(Sub-totals used for the ingredients)** | **Intervention running cost NGN (without investments)** |
| 1 | **Programme administration and staff cost** | | |
| | **cost of full time staff Administration** | | |
| | Stationary | 39,850 | 39,850 |
| | Printing and photocopies | 14,500 | 14,500 |
| 2 | **Targeting cost** | - | - |
| | Advocacy visits to 8 LGAs and government offices/agencies | 224,000 | 224,000 |
| 3 | **Staff training** | - | - |
| | **TOT workshop to review intervention messages/training** | - | - |
| | Refreshments/Lunches during training | 235,735 | 235,735 |
| | Printing of training materials | 10,500 | 10,500 |
| | Workshop materials | 8,500 | 8,500 |
| | Pre-testing of training tool | 143,850 | - |
| 4 | **Participants training** | - | - |
| | Participants teabreak and lunchfor 8 TRL training sessions(97 TRLs per session) | 1,480,000 | 1,480,000 |
| 5 | **Implementation and program material cost** | - | - |
| | Consultancy services for the development of training materials | 1,500,000 | - |
| | Production of handbook on vaccination (leaders with a heart for vaccination-development/printing),other graphics/flash cards | 8,200 | 8,200 |
| | Flash cards* | 143,531 | - |
| | Workshop materials | 35,330 | 35,330 |
| | Production of dashboard | 165,000 | - |
| | Transportation for trainers | 183,100 | 183,100 |
| | Communication for trainers | 36,000 | 36,000 |
| | **Healthworkers' training** | - | - |
| | Defaulters' register | 7,200 | 7,200 |
| 6 | **User costs** | - | - |
| | Opportunity cost of TRLs' time | 579,600 | 579,600 |
| 7 | **Averted cost** | - | - |
| | Cost of care for Measles | 186,620 | 186,620 |
| 8 | **Monitoring costs** | | |
| | Costs incurred by field staff for monitoring WDC meetings | 24,000 | 24,000 |
| | **Grand total** | **4,652,276** | **2,699,895** |
| | **Discounted to 2019** | **5,929,090** | **3,254,075** |

*4-year life annual costs

develop the training materials, and the production of the dashboard. The estimate of the marginal cost needed to scale up the intervention in additional adjacent wards in the future was made recognizing that the initial investments made to get the TRL program running in the initial sites would not be needed for the scale-up sites.

The cost was calculated using a full costing approach that included all the initial investments made to get the TRL project running but excluding the costs of managing the research project. An incremental cost-effectiveness analysis was conducted. For the actual calculation of the unit cost, the total intervention costs were divided by the net effect of the intervention for a series of outcomes. The incremental costs of the interventions compared to no intervention

(controls) were calculated. Similarly, the estimated total marginal cost was divided by these same net effect values to estimate the additional cost per unit of effect obtained if the intervention were to be implemented in additional wards in the future whereby the investment costs would not need to be repeated. The costs and outcomes were discounted according to the annual 10-year treasury bond rate of the Central Bank of Nigeria.

## Results

In the base case model, the totals costs amounted to 4,652,276NGN (15,189 US$). The average and marginal costs per ward were 387,690 NGN and 224,991 NGN, respectively. However, after discounting to 2019, the total, average and marginal costs were 5,929,090 NGN (19537 US$), 394,221NGN (1287 US$) and 216,361NGN (706 US$) (Table 4).

The estimate cost for averting measles illness was obtained from an earlier study based on the forecasted short-term cost that would be incurred for treatment of measles illness (i.e. cost of treatment, transport, caretaker lost wages) divided by the number of individuals with measles disease who likely sought health-care treatment [19]. In the study the different parameters were derived from country-specific estimates of proportions of children for whom care was sought for measles as well as data on the duration and rates of hospital admission while the estimated cost were primarily obtained from the WHO's Choosing Interventions that are Cost-Effective (WHO-CHOICE) project for the countries. This was estimated to be 7 US$ per care-seeking case averted [19].

The number of measles cases averted in our study was estimated based on the number of cases of measles reported by the respondents during the survey.

### Incremental costs and outcomes

In Table 5, the incremental unit costs for total cost and for running (marginal) cost are summarized. The first set of unit costs reflects the costs per unit of net effect to replicate the interventions in an entirely new setting. The second set of unit costs reflects the estimated cost per unit of net expected benefit in a setting where the initial investments are not necessary, in this case an adjacent ward.

**Table 4. Real and discounted cost of implementing TRL program and its scale-up.**

| 1 | Base Year: 2019 | Cost (NGN) | Cost (US $) | Discounted | Discounted cost to 2019 (NGN) | Discounted to 2019 (US$) |
|---|---|---|---|---|---|---|
| 2 | Total program cost for 12 wards | 4,652,276 | 15,189 | Total program cost for 15 wards | 5,929,090 | 19,357 |
| 3 | Average cost per ward*** | 387,690 | 1,266 | Average cost per ward*** | 394,221 | 1,287 |
| 4 | Average cost per 1276 children who could benefit ***** | 3,646 | 12 | Average cost per 1579 children who could benefit | 3,708 | 12 |
| 5 | Marginal cost to scale up to a new ward***** | 224,991 | 735 | | 216,361 | 706 |

*Exchange rate information*: 306.3 per USD******

* *This includes administrative cost, targeting cost, cost of developing and printing of the training tools, staff training, implementation, monitoring cost, and user cost.*

**Number of wards in which the intervention was carried out.*

****Average cost per ward.*

*****Average cost per eligible child in the ward.*

******This the marginal cost of adding one adjacent ward. The cost of developing the training tools and advocacy visit to communities were removed because these were oneoff activities. Also removed is the monitoring of community meetings as this was done by the Ward Focal person.*

******Bank rate: This was accessed on 23.05.2019 to establish the exchange rate on November 30*th *2018 being the end of the month project interventions were completed* [18].

**Table 5. Incremental cost-effectiveness ratio (ICER).**

| | | Costs (US$) per unit of outcome | | | Discounted Costs (US$) per discounted unit of outcome | |
|---|---|---|---|---|---|---|
| | Net effect (numbers) | Full cost per unit | Running cost per unit | Net effect (numbers) | Full cost per unit | Running cost per unit |
| **Cost of intervention** | | **15,189** | **8,815** | | **19,357** | **10,624** |
| Unit cost per ward | 12 | 1,266 | 735 | 15 | 1,287 | 706 |
| Unit cost per TRL trained | 97 | 157 | 91 | 122 | 159 | 87 |
| Absolute value for reduction of unvaccinated | 83 | 183 | 106 | 104 | 186 | 102 |
| Timely vaccination for Penta 1 | 197 | 77 | 45 | 247 | 78 | 43 |
| Timely vaccination for Penta 3 | 91 | 167 | 97 | 114 | 170 | 93 |
| Timely vaccination for measles | 129 | 118 | 68 | 162 | 119 | 66 |
| Number of Measles cases averted | 258 | 59 | 34 | 323 | 60 | 33 |
| Number of children who could benefit | 1276 | 12 | 7 | 1599 | 12 | 7 |

All outcomes showed net beneficial effects. The net effect also provides an approximate idea of 'how hard' it is to achieve good outcomes; for example, the net effect of timely Penta 1 is more than twice the net effect of Penta 3; suggesting that the latter may be more difficult to achieve.

The cost of the intervention per ward is an average based on a random sample of wards. It is difficult to estimate any economies of scale for implementing in larger wards without further collection of data. The average cost per TRL trained (about US$ 160) for the intervention gives another indication for estimating the cost of implementation on a broader scale: number of TRLs x US$ 160 for a ball park figure for the replication in an entirely new setting. Interestingly, the results show that the incremental discounted cost for every case of measles averted is only about 60 US$.

## Discussion

This study fills the knowledge gap about the economic evaluation of community-based interventions engaging traditional and religious leaders to improve immunization uptake in a low- and middle- income country setting. The analysis was based on primary data obtained from outcomes of a cluster randomized trial of the intervention and the cost incurred in running the program interventions. The effect of the intervention was estimated using logistic regression including random effects to adjust for non-independence of clustered observations at village, ward and LGA level.

The findings showed that the total cost of the intervention for the eight wards where this was carried out was US$15,189 giving a unit cost of US$12 for every child covered within the wards. When discounted, it showed that the total cost of the intervention for the eight wards was US$19,357 giving a unit cost of US$12 for every child covered within the wards. This study has shown that for every US$59 invested in this type of intervention one case of measles is averted in the intervention group compared to the control arm. However, for subsequent scale-up to new wards not requiring some of the initial investments made, the cost of averting one more case of measles drops to US$34. Similarly, it revealed that it will cost US$102 to have one more child fully vaccinated in the intervention group. With this amount there will be a reduction by one of the number of children who have not been fully immunized.

Engaging and training Traditional and Religious Leaders (TRLs) to promote uptake of childhood immunization in these rural Nigerian communities appears to have been more cost effective compared to results of a rural Indian study which assessed the cost-effectiveness of

providing health information to mothers on immunization through the use of a health messaging system [20]. On the other hand, our study generated higher cost per measles case averted than a study carried out in Benin Republic which had a routine immunization cost of US$5 for every measles case averted [21]. The difference in the cost may be due to the initial capital made in training and development of the training materials in our study and, may also be partly due to assumptions applied in costing estimation for that study. Again the differences may not be unexpected given that our intervention was an add on to improve the efficiency of immunization uptake using the routine delivery methods compared to theirs which was comparing different strategies for immunization delivery.

The use of primary data collected from a cluster randomized controlled trial is a strength of this study. Also a full costing of what was spent running the project was possible, thereby minimizing estimation for most of the cost incurred in setting up and running the intervention. However, this study has some limitations which include our inability to separate the costs per component since these were conducted concurrently and any separation would be a mere estimate. The full total costs as well as the full running costs used in the calculation, provide the most conservative unit cost calculation. The aim of this study was to have an overview of the cost of the intervention as a whole package, rather than having costs of single components of the multifaceted intervention.

Similar settings (like what is obtainable in most low-and middle-income countries) may have similar impact of the intervention. The intervention is more likely to be impactful where the traditional and religious leadership are embedded in one system. Furthermore, even though the TRL in Cross River State are under the same functional relationship, there are large differences between TRL in different wards. So, we could expect to see similar results in areas that have different TRL setups. However, it is difficult to ascertain how effective each component of the intervention would be and the cost-effectiveness of implementing single components of the intervention.

## Conclusion

Our multi-faceted intervention is highly cost-effective given that it is far less than double the annual national income per capita for most low and middle-income countries which is the threshold below which most Governments are highly recommended to fund interventions [22]. This increases further the maximum benefits from immunization services.

Thus it is recommended, that countries with a higher level of informal governance structures to consider the use of this strategy in improving their immunization coverage especially in hard to reach and rural areas where such structures operate maximally and which tend to lag in immunization coverage compared to urban areas.

## Acknowledgments

We wish to acknowledge our trainers Mrs Felicia Undelikwo, Mrs Christiana Bassey, Mrs Edim Oqua, Late Mrs. Eme Etim, Mrs. Felicia Eyaba for their immense contribution in the development of the training manual and in training the traditional and religious leaders.

Our appreciation also goes to the University of Calabar Demographic Health Surveillance for their support in setting up the data collection platform, training and ensuring entry and data quality. We would wish to acknowledge specifically Dr. Iwara Arikpo and his team: Anthony Okoro, Mboto Ekinya, Moses Bernard, and Eno Akan and for their painstaking efforts to validate the qualitative data.

## Author Contributions

**Conceptualization:** Angela E. Oyo-Ita, Xavier Bosch-Capblanch, Martin Meremikwu.

**Data curation:** Ogonna Nwankwo, Dachi Arikpo.

**Formal analysis:** Patrick Hanlon.

**Funding acquisition:** Angela E. Oyo-Ita, Xavier Bosch-Capblanch.

**Investigation:** Angela E. Oyo-Ita, Xavier Bosch-Capblanch, Dachi Arikpo, Ekperonne Esu.

**Methodology:** Patrick Hanlon, Ogonna Nwankwo, Xavier Bosch-Capblanch, Martin Meremikwu.

**Supervision:** Angela E. Oyo-Ita, Ekperonne Esu.

**Validation:** Angela E. Oyo-Ita, Dachi Arikpo, Ekperonne Esu.

**Writing – original draft:** Angela E. Oyo-Ita, Patrick Hanlon, Ogonna Nwankwo, Xavier Bosch-Capblanch, Christian Auer.

**Writing – review & editing:** Angela E. Oyo-Ita, Patrick Hanlon, Ogonna Nwankwo, Xavier Bosch-Capblanch, Dachi Arikpo, Ekperonne Esu, Christian Auer, Martin Meremikwu.

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
