## [Decision Letter · Decision Letter 0]

24 Feb 2021

PONE-D-20-25502

Cost-effectiveness analysis of an intervention project engaging Traditional and Religious Leaders to improve uptake of childhood immunization in southern Nigeria

PLOS ONE

Dear Dr. Oyo-Ita,

Thank you for submitting your manuscript to PLOS ONE. After careful consideration, we feel that it has merit but does not fully meet PLOS ONE’s publication criteria as it currently stands. Therefore, we invite you to submit a revised version of the manuscript that addresses the points raised during the review process. Please submit your revised manuscript by Apr 05 2021 11:59PM. If you will need more time than this to complete your revisions, please reply to this message or contact the journal office at plosone@plos.org. Please include the following items when submitting your revised manuscript:

We look forward to receiving your revised manuscript.

Kind regards,

Khin Thet Wai, MBBS, MPH, MA (Population & Family Planning Res.)

Academic Editor

PLOS ONE

Journal Requirements:

2. Please ensure you have included the registration number for the clinical trial referenced in the manuscript.

4. We note you have included a table to which you do not refer in the text of your manuscript. Please ensure that you refer to Table 4 in your text; if accepted, production will need this reference to link the reader to the Table.

Additional Editor Comments:

To improve scientific integrity, authors should focus on thorough revision of major methodological issues.

Reviewers' comments:

Reviewer's Responses to Questions

**Comments to the Author**

1. Is the manuscript technically sound, and do the data support the conclusions?

Reviewer #1: Partly

Reviewer #2: Yes

2. Has the statistical analysis been performed appropriately and rigorously? 

Reviewer #1: No

Reviewer #2: N/A

3. Have the authors made all data underlying the findings in their manuscript fully available?

Reviewer #1: Yes

Reviewer #2: No

4. Is the manuscript presented in an intelligible fashion and written in standard English?

Reviewer #1: Yes

Reviewer #2: Yes

5. Review Comments to the Author

Reviewer #1: The study objective is to examine the effectiveness and cost-effectiveness of using a multi-faceted intervention involving traditional and religious leaders for community engagement to improve uptake of routine immunisation services in Southern Nigeria.

As a cluster-randomised controlled clinical trial, there are a number of major methodological issues:

1.The authors had not elaborated the calculation and justification of the sample size estimation basing on a suitable anticipated effect size for the cluster randomization design.

2.Although LGAs and wards were said to be randomly selected, the process of randomization has not been clearly explained.

3.How was the outcome of vaccine uptake rate being ascertained in the participating villages? Were those routinely collected data, or collected and ascertainment with a specific procedure in this study? How was the variability of the completeness and accuracy of the data of ascertainment in different villages? In particular, would there be a differential degree of outcome ascertainment between intervention and control villages?

4.Many of the outcomes were not clearly defined and elaborated on how they were ascertained. This included how “timely vaccination” was defined, and how was “Number of Measles cases averted” and “Number of children who could benefit” be ascertained?

5.Was the study method, especially the presence of a control group being know by all participating villages? Would there be any bias on the observed outcome in either direction as a result of the awareness?

6.How much may the better vaccination uptake be attributed to a potential Hawthorne effect?

7.The method of statistical analysis of data has not been explained.

8.The calculation of the counterfactual effect as reported on Table 2 has not been clearly explained. Is that just a simple subtraction of the effect between the intervention and control group? The figures does not exactly adds up.

9.Can the authors also discuss on the extent of external validity of the result, including both the nature of the (or a similar) intervention, and the observed clinical effectiveness and cost-effectiveness of the intervention, to other villages and areas of Nigeria or countries with a comparable degree of socio-economic development, or to other more developed areas/ countries.

Reviewer #2: The authors have carried out an economic evaluation of a community-based interventions engaging traditional and religious leaders to improve immunization uptake in a state of Nigeria. it is good that the cost data was collated during the execution of a cluster randomized trial.

The following lines were unclear or could do with some explanation:

1.Line 101 point 2 on community engagement does not make sense: "Routine community for a were used by the TRLs to share information on routine immunization "

2.It would be useful if the authors could expand on lines 170-172 on how they estimated cost of measles illness "The estimate for averting measles was estimated based on the estimated short term cost of measles illness (i.e. cost of treatment, transport, caretaker lost wages). This was estimated to be7 US$ per care-seeking case averted"

3.Likewise it would be useful if they elaborated on how they estimate number of measles cases averted? (lines 172-74) "estimated based on the number of cases of measles reported on the District Health Information System."

4.Re: line 245-48: “The findings showed that the total cost of the intervention for the eight wards where this was carried out was US$15,189 giving a unit cost of US$12 for every child covered within the wards”. Was there variation in unit costs between different wards? While the aggregate number is useful for deriving a unit cost, it would be interesting to know if unit cost varied between wards and if so how much?

5.Line 253: "Similarly, it revealed that to have one more child to be vaccinated with all the vaccine antigens including measles antigen on time in the intervention group compared to the control group US$102 will need to be spent." it would be useful if the authors explained this.

6.Line 283: "Thus it is recommended, especially in low and middle income countries that have a higher level of informal governance structures to consider the use of this strategy in improving their immunization coverage especially in hard to reach and rural areas where such structures operate maximally and which tend to lag in immunization coverage compared to urban areas."

The evidence pertains to the context of Nigeria but is it appropriate to make a sweeping generalization to LMICs? Considering the study was carried out in one of 36 states of Nigeria.

6. PLOS authors have the option to publish the peer review history of their article (what does this mean?). If published, this will include your full peer review and any attached files.

Reviewer #1: No

Reviewer #2: No

---

## [Author Response · Author response to Decision Letter 0]

3 Jun 2021

RESPONSE: Done

2. Please ensure you have included the registration number for the clinical trial referenced in the manuscript.

The registration number of the trial is recorded in the referenced effectiveness paper

RESPONSE: The data for the effectiveness study is uploaded in Dryad. The DOI for the data is: https://doi.org/10.

5061/dryad.gmsbcc2ms. Cost related data were extracted from trial records and presented in the manuscript

4. We note you have included a table to which you do not refer in the text of your manuscript. Please ensure that you refer to Table 4 in your text; if accepted, production will need this reference to link the reader to the Table.

 RESPONSE: Reference has been made to Table 4

Response to reviewers' comments is uploaded.

---

## [Editor Report · Decision Letter 1]

31 Aug 2021

Cost-effectiveness analysis of an intervention project engaging Traditional and Religious Leaders to improve uptake of childhood immunization in southern Nigeria

PONE-D-20-25502R1

Dear Dr. Oyo-Ita,

We’re pleased to inform you that your manuscript has been judged scientifically suitable for publication and will be formally accepted for publication once it meets all outstanding technical requirements.

Kind regards,

Khin Thet Wai, MBBS, MPH, MA

Academic Editor

PLOS ONE
---

## [Editor Report · Acceptance letter]

7 Sep 2021

PONE-D-20-25502R1 

Cost-effectiveness analysis of an intervention project engaging Traditional and Religious Leaders to improve uptake of childhood immunization in southern Nigeria 

Dear Dr. Oyo-Ita:

I'm pleased to inform you that your manuscript has been deemed suitable for publication in PLOS ONE. Congratulations! Your manuscript is now with our production department. 

Kind regards, 

on behalf of

Dr. Khin Thet Wai 

Academic Editor

PLOS ONE